# Establishment and Application of Crowd-Sensing-Based System for Bridge Structural Crack Detection

Hangming Yuan [1,2], Tao Jin [2,3,*] and Xiaowei Ye [2]

1    Polytechnic Institute, Zhejiang University, Hangzhou 310058, China
2    Department of Civil Engineering, Zhejiang University, Hangzhou 310058, China
3    School of Engineering, Hangzhou City University, Hangzhou 310015, China
*    Correspondence: cetaojin@zju.edu.cn

**Abstract:** The inspection of bridge structural cracks is essential to the structural safety evaluation and could provide reference for preventive maintenance. The traditional bridge structure inspection methods rely heavily on trained engineers with professional equipment. While such kind of way could provide reliable crack inspection data, the enormous amount of existing bridges waiting for inspection challenges the efficiency of these methods. Fortunately, the development of smartphones facilitates the possibility of making the pedestrian taking smartphones a mobile sensing node, which is able to collect crack information such as images and locations. At the same time, the booming deep learning methods could offer remarkable crack detection capacity to deal with the crack images automatically. Given this consideration, this paper established a crowd-sensing-based system for bridge structural crack detection. The system was composed of the cloud-based management platform and the mobile based application (APP) for crack information collection. The mobile-based APP was used by the volunteer pedestrians to collect the crack images as well as the locations, and the location accuracy was estimated to be around 5~10 m. Meanwhile, the cloud-based management platform was used for the management of the users and the collected crack information uploaded by all of the volunteers. A deep neural network was used to deal with the crack detection tasks and evaluate the quality of the collected images to see if they could be fitted for crack detection in bridge inspection works.

**Keywords:** structural health monitoring; crowd sensing; smartphone; deep neural network; crack detection





## 1. Introduction

As an important part of transportation infrastructure, the health condition, safety and durability of bridge structures are of great concern to industry, academia, and the general public, as they are related to safe travelling for residents and the formulation of infrastructure maintenance decisions [1,2]. Traditional bridge structure inspections are carried out by maintenance companies, which draw up inspection plans based on the distribution and quantity of bridges waiting for inspection [3–5]. On the one hand, with the development of China's economic development, the traffic flow on the roads is growing. The number of vehicles and their weights are constantly increasing, causing greater pressure on the structural safety of the existing bridges. On the other hand, as China's transportation infrastructure continues to develop, the number of bridge constructions is rapidly increasing, and the distribution area of bridges is also gradually expanding, requiring more maintenance forces to carry out bridge maintenance [6–9]. However, the number of specialized equipment for bridge inspections, such as bridge inspection vehicles and elevated work vehicles as well as professional inspectors, is insufficient, and the cost of labor is also increasing [10,11]. Therefore, the insufficient detection force is unable to meet the constantly increasing demand for bridge inspection. In the case of limited force, the

adjustments that the inspection units may need to make include prolonging the inspection cycle for bridges or using the limited force to inspect large bridges or critical routes, reducing the inspection of bridges on minor traffic routes. Failure to conduct timely inspections will have adverse effects on the safety and durability of bridges [12–15].

Early crack detection is extremely important because cracks at this stage are fine in width and short in length, but as cracks expand, they can lead to structural failure [16,17]. In the past, many crack detection and characterization techniques have been developed through research and development by experts in the fields of SHM [18,19] (Structural Health Monitoring), NDE [20] (Non-Destructive Evaluation), etc. Duan et al. [21] proposed a CNN-based damage identification method for hanger cables in a tied arch bridge and the result showed that the CNN trained by Fourier amplitude spectra was better than the CNN trained by time history data. Mangalathu et al. [22] used a CNN to establish a prediction model for evaluating ground motion and structural damage state. The training data were obtained from numerical simulating. Guo et al. [23] used the CNN algorithm and designed a multi-scale module which could extract damage features from noisy and incomplete mode shape data. Measuring these temporal and spatial changes through traditional methods such as field observation or surveys is quite a challenging and time-consuming task [24]. Therefore, it is encouraged to explore new technologies for more efficient inspection methods, such as the utilization of smartphones with Global Position System (GPS), in order to reduce costs and increase participation of the public pedestrians [25–27].

Crowd sensing is a concept that emerged from the development of the Internet of Things and is a new way of acquiring monitoring data [28]. Crowd sensing can efficiently and cost-effectively collect a massive amount of heterogeneous data through portable mobile devices carried by a large number of end users on the mobile end [29–31]. Currently, smartphones equipped with various sensors such as image sensors, GPS sensors, and gyroscopes have become one of the basic items carried by the general public. The development of mobile networks such as 4G and 5G ensures that remote mountain areas can also achieve fast transmission of files, including images and even videos [32–37]. Due to the demand for work, life, or entertainment, thousands of citizens naturally need to travel on various bridges [38,39]. Therefore, the general public carrying smartphones is, in fact, a mobile sensing node that can use the sensors on the phones to record and transmit crack image information and corresponding spatial locations at high frequencies [40]. The traditional inspection process uses a camera to take pictures of cracks, record the location and damage description on papers, and return from the field for post-processing. Meanwhile, crowd sensing with smartphones can directly record and upload crack images and location information, which is more advantageous for digitizing and automating the recording of crack information and simplifying data processing [41–43].

Based on the idea of crowd sensing and the characteristics of bridge crack collection, this paper established a crowd-sensing-based bridge structural crack detection system. A crowd sensing crack information collection system was developed, which contains a cloud management platform as the center and the user mobile application as the sensing node. A deep neural network was adopted for the detection of cracks in the uploaded images. The investigation can provide reference for the establishment of crowd-sensing- and deep-learning-based system for crack inspection of bridge structures.

## 2. Crowd-Sensing-Based Crack Detection Method

After detecting cracks on a bridge, it is necessary to reposition them for timely repair. Therefore, the location information of bridge cracks is crucial for maintenance work. Due to the sparse distribution and vast coverage of bridge structures, crack positioning presents significant challenges. The traditional positioning method involves manually recording on papers during field inspection, and then manually inputting the data into a computer after the field inspection process. This process has low digitization and automation, and with an increase in the number of cracks, manual inputting can lead to recording errors. Moreover, due to the lack of a uniform coordinate system, the recording of crack locations

is subjectively categorized by inspectors based on bridge direction or pier numbers. Such methods have low standardization and poor readability, which makes the process of crack data recording and transmission difficult. In this section, based on image acquisition using crowd sensing, a crack positioning method based on the GPS system was proposed.

The GPS is a satellite-based navigation and positioning system designed by Brad Parkinson and his team in the United States after 20 years of effort. It is capable of accurately positioning most areas on the Earth's surface [44,45]. The GPS has already been applied in civil engineering for bridge displacement monitoring, building structure deformation, dam displacement, etc. [46,47].

The crowd sensing crack collection system requires participants to collect data on their mobile devices and then gather the information on the cloud for analysis and processing. The design concept is shown in Figure 1. The hardware of the system consists of four parts, including the application installed on the participants' mobile phones, the transmission network represented by signal towers and broadband networks, the cloud server for data storage and analysis, and the management unit's terminal computers.

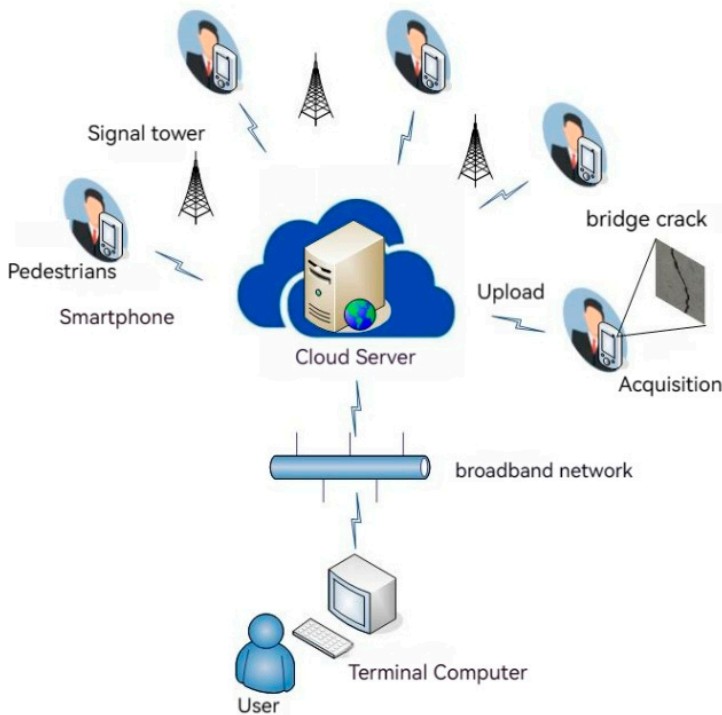

**Figure 1.** Components of the crowd sensing crack collection system.

## 3. Development of the Crowd Sensing Crack Collection System

The transmission network in the crowd sensing crack collection system shown in Figure 1 can utilize thd existing broadband and mobile network facilities, as smartphones have become almost ubiquitous mobile devices nowadays. What needs to be built is the cloud server, the cloud-based mobile APP management platform, and the mobile APP installed on mobile phones. The cloud server can be rented with selected configurations according to the needs. The architecture of the mobile APP and the management platform in the cloud server are shown in Figure 2. The main functions of the cloud-based management platform contain participant account management, crack information management, and crack information inquiry. The main functions of the mobile APP contain crack image capturing, location recording, text description of cracks, and record inquiry.

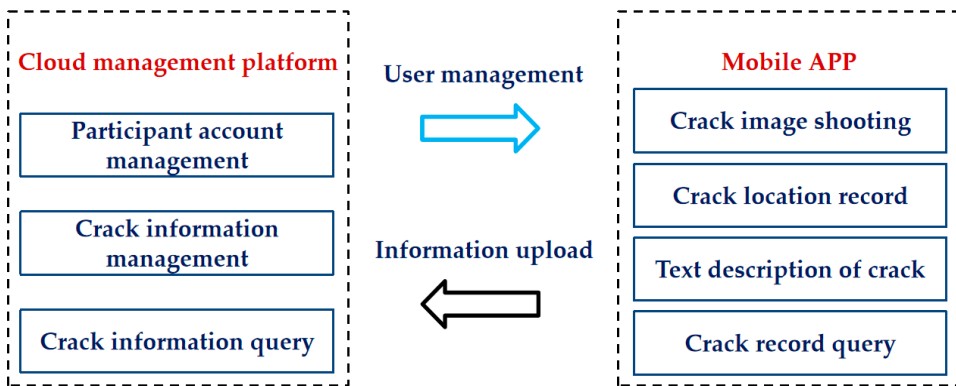

**Figure 2.** Architecture of the crowd sensing crack collection system.

The crowd sensing approach adopted by this system is the active sensing by people in the loop. Widely distributed participants carrying smartphones will come close to bridge structures such as railings, road surfaces, beams, and bridge piers in their daily travels. Participants can use the mobile APP to capture crack images; meanwhile, the APP will record the locations of the participants at the time and upload them to the cloud server. The users can process the collected data through the management platform in the cloud server by logging in via the web.

### 3.1. Cloud Server Configuration

The trend of big data and cloud computing in recent years has led to the construction of cloud servers. Cloud servers can achieve centralized processing, which is beneficial for data maintenance and management. The crowd sensing crack collection system collects cracks through participants, and the centralized processing of the cloud server is conducive to user management, crack information processing, and utilization, providing decision-making support for bridge maintenance-related units. The issues mainly involved in cloud server configuration are the selection of the cloud platform and the hardware configuration of the server.

#### 3.1.1. Cloud Platform Selection

There are currently two operating modes for cloud platforms: private cloud platform and public cloud platform. Private cloud is mainly built for individual users. Building a private cloud platform requires purchasing dedicated servers and configuring auxiliary measures such as management rooms, power supply, and cooling system to ensure its normal operation. Since it is self-built, it has the advantages of ensuring data security and effective service quality. However, private cloud platforms require purchasing hardware to meet the system's expansion needs, configuring dedicated technical personnel for maintenance and operation and requiring extra resources for cooling and other auxiliary measures. Therefore, the cost of private clouds is high, which is not conducive to development testing and large-scale popularity.

Public cloud platforms are cloud platforms built on cloud services provided by third-party vendors. Users can access them via the internet and configure and purchase according to their needs. The main advantage of public clouds is the convenience for data sharing and scalability. Users do not need to configure dedicated computer rooms, networks, IP addresses, cooling equipment, or technical managers. Instead, they can pay on-demand and receive virtual private cloud services. Users distributed in various regions can quickly access them via the internet, making data sharing and management convenient. As the data volume or processing requirement increases, public cloud platforms can be flexibly upgraded for data storage and processing. The expansion potential of storage space and computing power is quite huge.

Common vendors in the current market for cloud service platforms include Amazon Web Services, Microsoft Azure, Alibaba Cloud, and Huawei Cloud. For the deployment of cloud servers in this section, Alibaba Cloud has been chosen as the cloud platform. Founded in 2009, Alibaba Cloud has rich and mature service experience in fields such as manufacturing, financial services, government services, traffic monitoring, healthcare services, telecommunications services, and energy supply. Therefore, in this section, Alibaba Cloud is used as the public cloud platform for deploying the cloud server.

### 3.1.2. Server Hardware and Software Configuration

The hardware and software configuration of the server deployed on Alibaba Cloud is shown in Table 1. As this system is currently in the experimental stage, the rented configuration is relatively basic, which is enough to meet the functional requirement. In the future, the hardware and software configurations can be upgraded to meet larger computing and storage requirement.

**Table 1.** Server hardware and software configuration.

| Overall | Sub-Category | Version |
| --- | --- | --- |
| Hardware | Memory | 2 G |
| | Hard Drive | 40 G |
| | Bandwidth | 1 M/s |
| Software | Operating system | Linux cent Os 7 |
| | Database | MySQL 5.7 |

### *3.2. Development of Cloud Management Platform*

The cloud management platform is the core of the crowd sensing crack collection system, responsible for account management, crack information management, and has the query function to check and display the recorded crack information. The management platform is divided into front-end and back-end. The front-end is developed using Vue.js and combined with third-party components of the Gaode Map. The back-end is developed using Java.

### 3.2.1. Function Design

The functions of the cloud management platform are as follows: (i) User login: The users of this platform are bridge maintenance management units, and the number of users is customized by the backend. Users can remotely log in from any computer with internet access using their account and password; (ii) Participant account management: As the target participants of this system are daily commuters and the participation is voluntary, the system needs to assign accounts for those who are interested in this work. When participants are no longer involved, their account can be deleted to release the corresponding resources; (iii) Crack information management: Crack information includes images, GPS coordinates, and the corresponding text descriptions. Crack information management allows users to download images, coordinates, and texts individually or in batches for use; (iv) Crack information search: As crack collection is a long-term project and many images can be uploaded daily, crack information search allows users to search crack images by time and can directly display the images on the map, enabling users to quickly obtain the crack positions.

### 3.2.2. Interface Design

For the convenience of users, this management platform realizes remote access through web login. Users can access it through various mainstream browsers such as the IE, the 360 browser, and the Google Chrome. The main login interface is shown in Figure 3. The interface of this cloud management system adopts account and password login. Initially, a set of account and password is set up for this system.

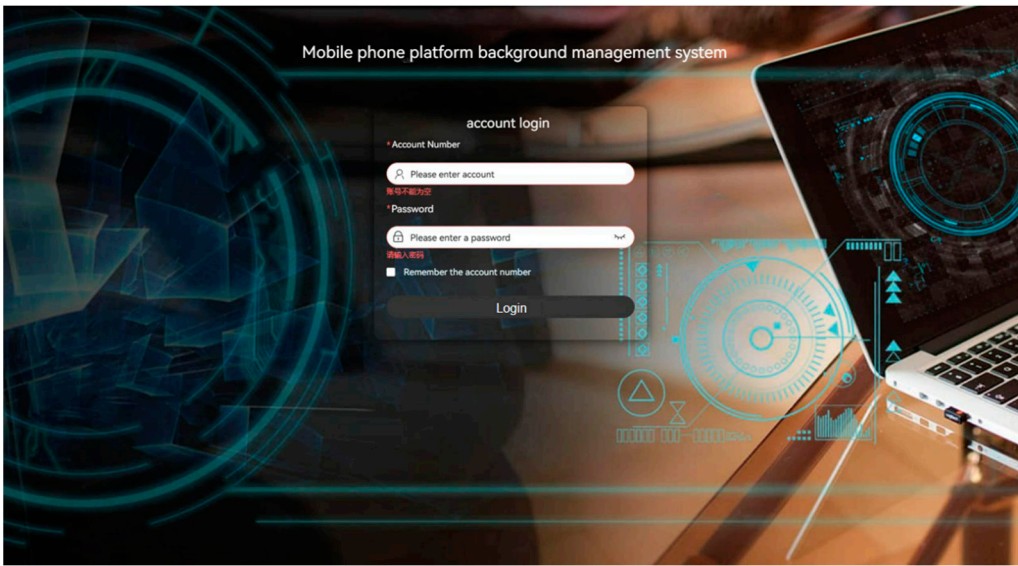

**Figure 3.** Cloud management platform login page.

As shown in Figure 4 above, after logging in, the user enters the main interface of the management platform. The main interface contains three modules: account management module, record management module, and record query module. In the account management module, the number of accounts, names, phone numbers, creation time, account numbers, and passwords can be determined. The operations that can be performed on the accounts include editing, deleting, and adding. The interface for adding a new user is shown in Figure 5 below.

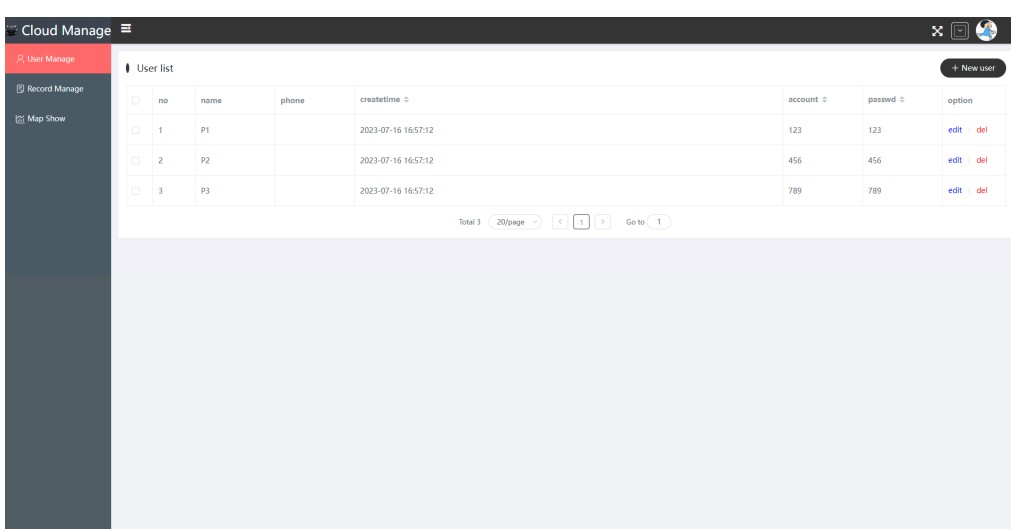

**Figure 4.** Cloud management platform main interface.

The new user module can be used to increase the number of users and enhance the ability of crack detection. Privacy protection is an important part of crowd sensing systems, which can encourage participants to register and participate in sensing tasks. In order to protect the privacy of the participants, the cloud management platform only requires an account and a password to log in, and the user's name and phone number are optional, which also preserves the incentive mechanisms for future task rewards and other research purposes.

**Figure 5.** Adding user interface.

As shown in Figure 6, the record management interface can display the total number of images, the reporters and their respective reporting times for each image. The GPS-recorded latitude and longitude coordinates in the crack image information can reflect the location of the crack on the map. The pitch angle, heading angle, and roll angle recorded by the gyroscope can record the posture of the phone when taking pictures, thus assisting in inferring the location of the crack. The crack thumbnails allow users to have a visual understanding of the crack status. The buttons on the rightmost column of the interface can allow users to download or delete a single image or read text explanations of the crack through the details option. For convenience, the upper right corner of the record interface sets a batch processing button, where users can choose to download multiple images or download location information and text explanations in bulk, or download all at once. The crack images are downloaded in JPEG format, while other information is downloaded in Excel format, making it easy to handle through third-party software such as Matlab 9.11. Clicking the details button takes the user to the text description box, shown in Figure 7.

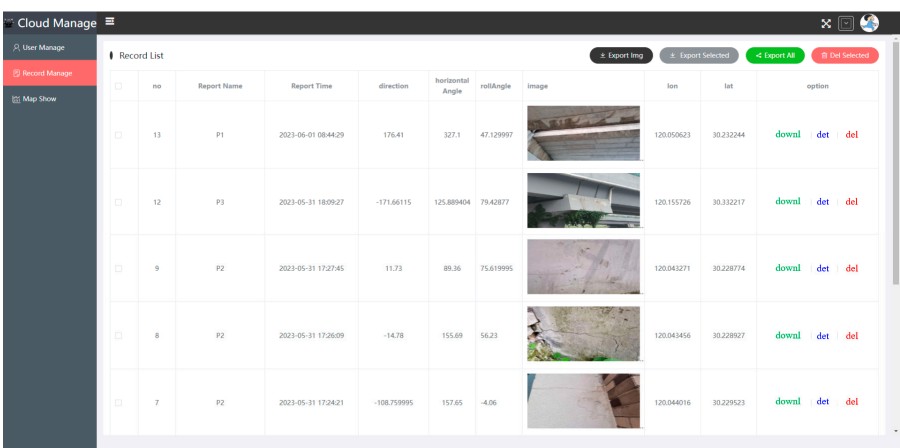

**Figure 6.** Record management interface.

Due to the complex and varied on-site situations, in addition to collecting crack images, and GPS coordinates, the system also retains the function of adding text remarks, as shown in Figure 7. Thus, participants can briefly describe the on-site situation and assist in determining the state of the bridge components or cracks.

As shown in Figure 8, in order to visually display the crack images recorded by pedestrians, the cloud-based management platform set up a recording inquiry module. This module can provide a search function based on the photographing time and directly

display the crack recording positions on the map. Clicking on the crack recording point can directly display the crack information, as shown in Figure 9. The cloud-based management platform can view and download a variety of heterogeneous data sources including the coordinates, crack images, and text descriptions of cracks in order to facilitate centralized data processing.

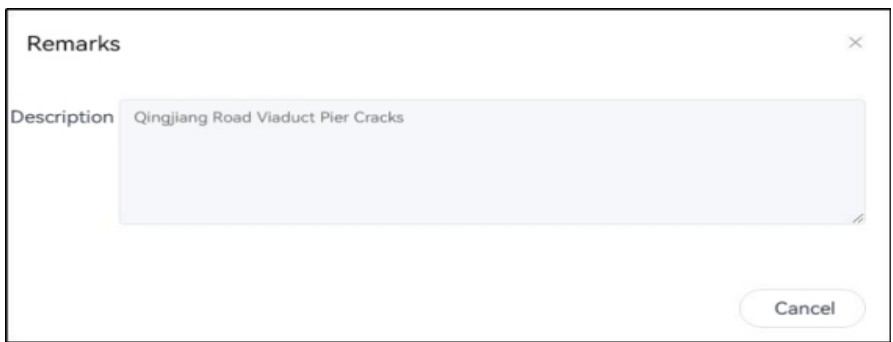

**Figure 7.** Crack text description.

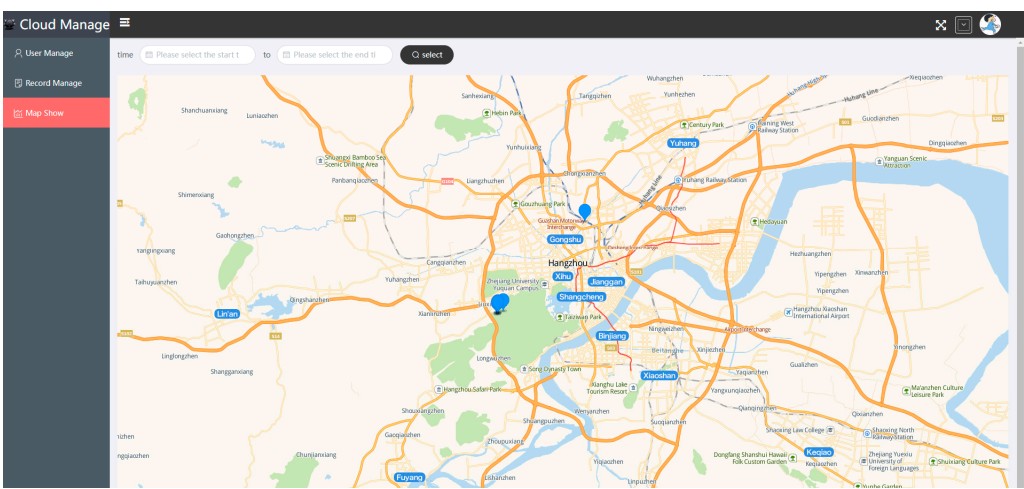

**Figure 8.** Crack record query.

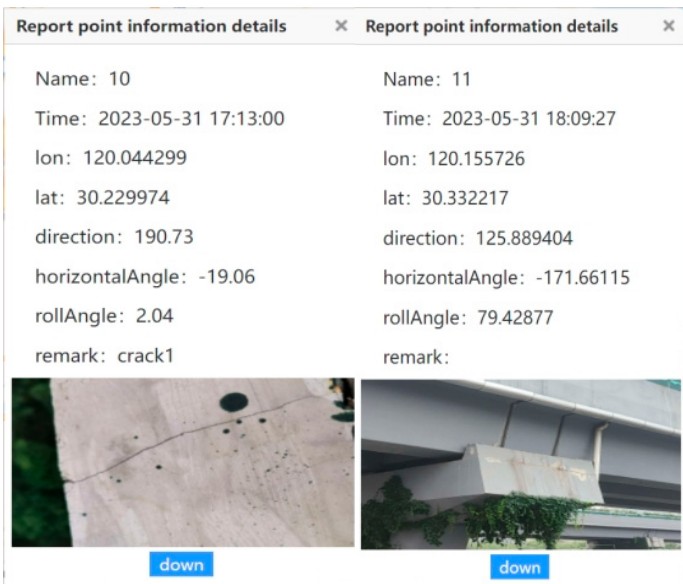

**Figure 9.** Single-crack information viewing demonstration.

### 3.3. Mobile App Development

The mobile application is developed based on the Android system, and the reason for choosing the Android system is its good open-source nature, large user base, and coverage of various phone models. An application based on the Android platform can reach a wider audience and contribute to the growth of the number of crack inspection users.

### 3.3.1. Function Design

The mobile application for participants is designed to allow them to record and upload information related to cracks. Therefore, its main functions are as follows: (i) Crack image acquisition: Participants can use the application to capture the cracks in real time by tapping the phone's camera to record the bridge cracks. The images can be saved separately in a folder. Also, while capturing the crack images, the mobile phone also uses the GPS sensor to record the location of the smartphone; (ii) Text description: A module is added to enable participants to describe the situation more accurately and provide more information on complicated and unforeseen issues that may arise on site; (iii) Record query: Participants can query their historical records for the crack images they have saved, which are used later for potential reward applications or other incentive measures.

Considering the extensive range of users who might use the application, it is designed to simplify the process and reduce the learning requirement of the participants. Therefore, the process of capturing cracks has been simplified to guide the participants through taking and uploading crack images in a step-by-step manner, as shown in Figure 10.

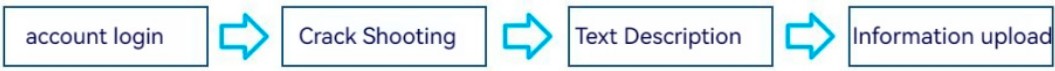

**Figure 10.** Crack information collection process.

### 3.3.2. Page Design

The mobile application can be installed directly through the generated APK file. Once installed, the Crack Detection Application can be accessed as shown in Figure 11a. After logging in with a username and password, users can access the main interface as shown in Figure 11b. The main interface of the application features a prominently displayed green button used to activate the camera for crack photographing. At the same time, the GPS sensor is also activated to record relevant location information.

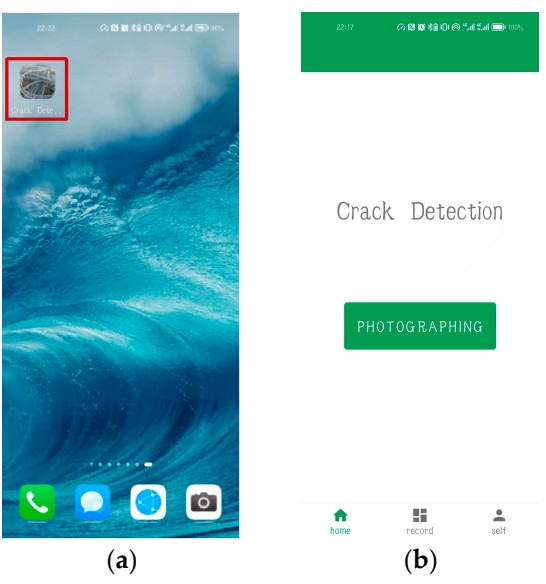

(a)                                    (b)

**Figure 11.** Crack collection application. (**a**) Mobile Appicon. (**b**) Main interface of the application.

After clicking on the photographing button and activating the camera, users are led to the image capturing interface shown in Figure 12. Figure 12a displays the real-time camera feed, and after capturing an image, users can view the results as shown in Figure 12b. The captured image displays a thumbnail of the crack, the GPS coordinates recorded, the phone's orientation recorded by the gyroscope, and a landmark nearest to the capture location based on the GPS coordinates on the map. If the captured image clearly displays the features of the crack, users can proceed to add a text description. Otherwise, users may choose to recapture the crack image.

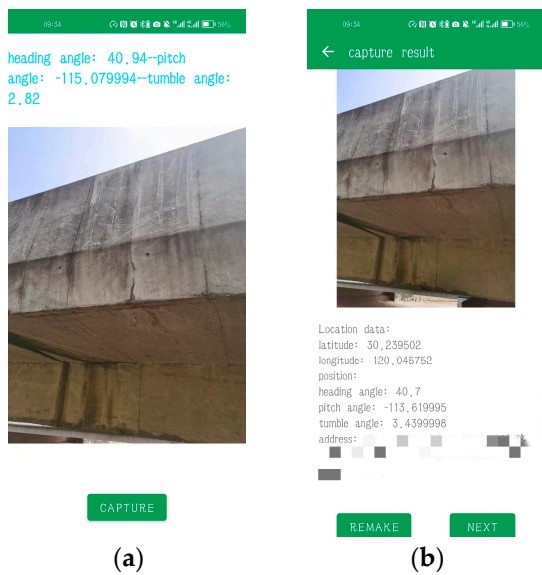

(**a**)        (**b**)

**Figure 12.** Crack image acquisition. (**a**) Real-time picturing. (**b**) Picturing result.

Users can input the text description directly using their device's input method. After completing the description, users can click on the upload button to upload the image, GPS coordinates, phone orientation data, and text description to the cloud management platform simultaneously.

There are two ways for participants to view the crack information they have captured. Firstly, they can click on "records" on the main interface. Secondly, they can directly view the crack photos stored in the phone folder. In the photo viewing mode, users can browse the crack images directly in the folder stored in the phone. In the record viewing mode, users can view all the information related to the crack and see its location on the map. Keeping records can show participants their past work and encourage them to take more initiative. It can also serve as one of the supporting documents for potential reward mechanisms.

## 4. Crack Detection Test

In order to test whether the crack images obtained from the crowd sensing system can achieve satisfactory recognition results with deep learning models, we conducted recognition experiments using crack images taken by users in different scenes and with different shooting devices. As for the position accuracy in this study, it was estimated to be around 5~10 m. Previous studies have conducted researches on the accuracy of smartphone- and GPS-based location method. In terms of location accuracy by smartphone without post-processing, Merry and Bettinger [48] obtained an accuracy range of 7~13 m, Zandbergen [49] reported an accuracy of around 10 m, Menard et al. [50] found an accuracy range of 5~10 m, Garnett and Stewart [51] pointed out an accuracy of around 6.5 m. Furthermore, a meter-level accuracy could be achieved by the establishment of a differential correction method [52], an improved Hatch filtering method [53], and the RTK-based methods [54,55].

Given this challenge in crack location, part of the reason for the crack description function shown in Figure 13 was to provide assistance for crack location description.

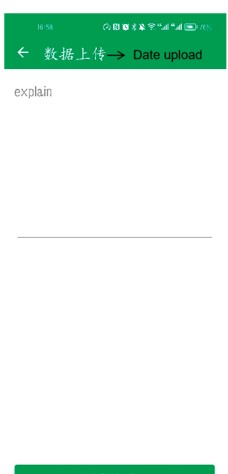

**Figure 13.** Crack description.

The uploaded user images have varying pixel sizes ranging from 4000 × 3000 to 2000 × 1000, and we randomly selected a few crack images for recognition. A U-Net-based model was trained to deal with the crack detection task for evaluation of the quality of the crack images captured in this way. The U-Net was proposed by Ronneberger et al. [48] for semantic segmentation. It has a U-shaped encode–decode architecture that can output the detection result in the same size with the input image. Thus, it is suitable to be trained for crack detection in a pixel-wise manner. The training process lasted for 200 epochs for the optimization of parameters in the network, and it was based on a publicly available dataset (published on 8 February 2021) at: https://doi.org/10.7910/DVN/RURXSH. The code and the trained model were uploaded to Harvard Dataverse on 31 May 2023, and it is publicly available at: https://doi.org/10.7910/DVN/IFMJLE.

Crack detection results of eight images with different crack geometries and backgrounds were adopted for illustration. The results of the actual image recognition using the deep learning model are shown in Figure 14. From the original images, it can be seen that the lighting conditions of mobile-phone-captured images are uneven, and different from the images used in the training process. Moreover, the large size of the images leads to the uneven distribution of pixels on the entire image, which is an important factor affecting the accuracy of recognition. Due to the large contrast between the crack and background noise, crack images can be recognized better in bright background images, such as the recognition results of Figure 14a–c in the figure below. The boundary between the background and crack is clear and the contrast is high, which enables the deep neural network model to achieve good recognition results. When the contrast between the background and crack is low, the model recognition is not accurate, error detection (marked with red boxes) happened, as shown in Figure 14d,e. In image e, part of the crack appears white due to the structural material, which causes the neural network to have obvious disconnection recognition result (marked with red boxes). Image d also has structural joint seams, which is a noise motif that is easily recognized as a crack in the network recognition. However, since the joint seams have an unclear difference in morphology from the regular cracks, the continuous and straight extension of the joint seams is easier to be identified as noise crack recognized by the model. The background colors of Figure 14f–h are relatively diverse compared to those of the above images, which leads to more erroneous noise obtained by the network during recognition. Especially, the thinner crack areas are more likely to be mixed with the surrounding background, resulting in recognition errors, as is shown in the red-box-marked areas in Figure 14f,h, which have obvious disconnections. Meanwhile, the crack detection of the network in image g achieved a satisfactory result.

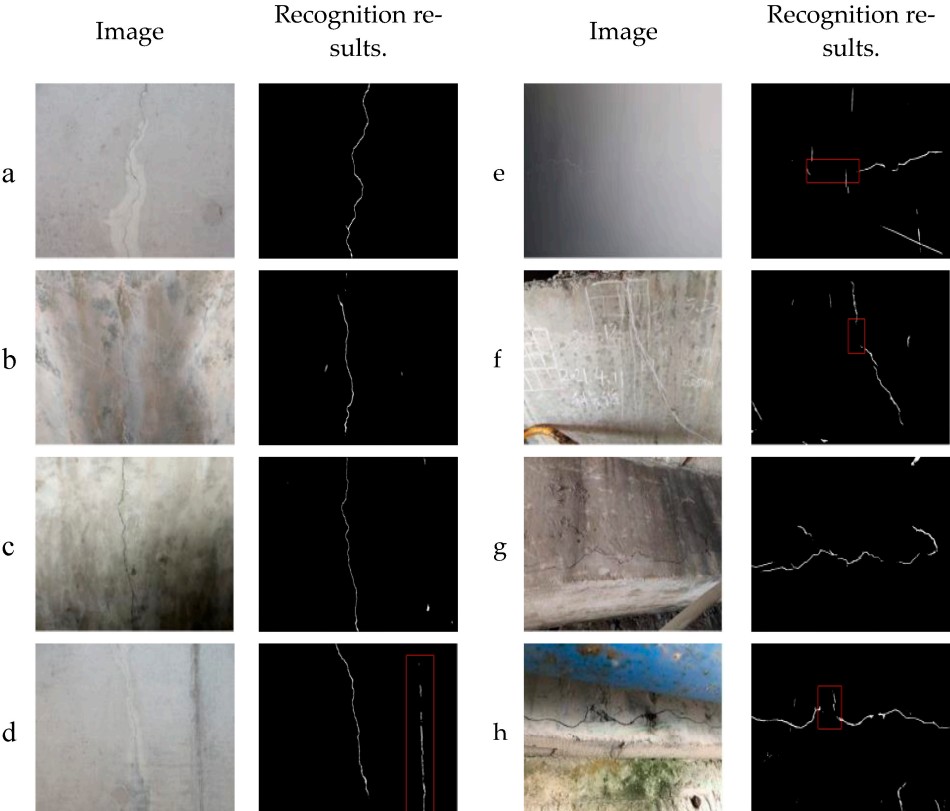

**Figure 14.** Crack detection result.

## 5. Conclusions

This paper established a crowd-sensing-based system for bridge structural crack detection aiming at facilitating the participation of pedestrians to help structural crack detection tasks. A deep neural network was utilized to deal with the collected structural crack images to evaluate whether those crack images were fitted for structural crack detection. Based on the investigation, some conclusions can be drawn as follows:

(i) The crowd-sensing-based system for bridge structural crack detection was consisted of the cloud-based management platform and the mobile-based APP for crack information collection. The cloud platform was designed to conduct user management, crack information management and record query tasks. Meanwhile, the mobile-based APP could capture the crack images, location information and allow for short description. The location accuracy was around 5~10 m, and the crack description function can be used to provide assistance for crack location.

(ii) The preliminary tests showed that the system could realize the intended crack information acquisition purposes. However, since these images were not obtained by trained engineers with uniform equipment, the quality, picturing condition and angles, and the resolution of the crack images varied.

(iii) The crack detection results based on the deep neural network and the smartphone-based images showed that the deep neural network could detect most of the crack regions on these images when the background was clear and bright. Meanwhile, when the background was less contrast and complex, noise motifs could cheat the deep neural network.

**Author Contributions:** Conceptualization, T.J. and X.Y.; methodology, T.J. and H.Y.; validation, X.Y. and H.Y.; formal analysis, H.Y.; investigation, T.J. and X.Y.; resources, X.Y. and H.Y.; data curation, H.Y.; writing—original draft preparation, H.Y. and T.J.; writing—review and editing, X.Y. and T.J.; visualization, X.Y.; supervision, T.J.; project administration, X.Y.; funding acquisition, T.J. and X.Y. All authors have read and agreed to the published version of the manuscript.

**Funding:** The work described in this paper was jointly supported by the China Postdoctoral Science Foundation (Grant No. 2022M712787), and the National Natural Science Foundation of China (Grant No. 52178306).

**Institutional Review Board Statement:** Not applicable.

**Informed Consent Statement:** Not applicable.

**Data Availability Statement:** The data that support the findings of this study are available from the corresponding author upon reasonable request.

**Conflicts of Interest:** The authors declare no conflict of interest.

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
