# Peer review of "Establishment and Application of Crowd-Sensing-Based System for Bridge Structural Crack Detection"

_applsci, doi:10.3390/app13148281_

Round 1
Reviewer 1 Report
The manuscript presents a crowd-sensing based system for bridge structural crack detection. The system consisted of the cloud-based management platform and the mobile application for crack information collection.
The concept of manuscript is very interesting and has several potentials for actual applications.
I think the manuscript is in good shape. Kindly find below some comments that can help improving it.
Comments:
- Abstract: Please rewrite the following “With the development of smartphones, the general public carrying smartphones is literally a mobile sensing node that can record crack image in formation and locations.”
- Abstract: No need for abbreviation “APP”. Use application. Or at lease define it first.
- Figure 2.1 name , revise “composition” to “components”.
- Figure 3.1, change font type to match manuscript font type.
- Language needs to be checked for fluency and grammar. Several mistakes.
check for fluency and grammar is required.
Reviewer 2 Report
This paper presents an interesting work for bridge structural crack detection by using the concept of crowd-sensing, where pedestrians are involved in bridge structural health monitoring by using a smartphone. Deep learning models are used to detect the crack region based on those uploaded images. Before the acceptance, a few questions need to be addressed:
(1) For Section 4, crack detection can be achieved easily by using U-Net, or other similar algorithms as shown in Figure 4.1. What about crack localization by using a smartphone? More specifically, what’s the GPS accuracy of using a smartphone? Please conduct some investigations on the accuracy of crack localization by using a smartphone.
(2) Regarding the U-Net as described in Section 4, could you give more details about the U-Net algorithm, like its network structure, and detailed operations for each layer?
(3) What’s the current status of the proposed system in practice? Have you launched the system to the public? Or, is it still in the preliminary stage for research only?
Round 2
Reviewer 2 Report
In the Abstract and Conclusions, the authors need to point out/discuss the estimated accuracy in crack localization by using smartphones. Apparently, the localization error of 5-10 m (or even larger) causes significant uncertainty regarding the bridge SHM, especially for short-span bridges. Additionally, regarding the estimated accuracy in crack localization, the authors need to cite some relevant references regarding those mentioned localization accuracy if they have not investigated it themselves. If they have, please present the results in this manuscript.
